**Investigation**

# Genetic regulation of fasting-induced longevity effects

Alison Luciano, Laura Robinson, William H. Schott, A. Phillip West (iD), Ron Korstanje (iD), Gary A. Churchill (iD) *

The Jackson Laboratory, Bar Harbor, ME 04609, United States

*Corresponding author: The Jackson Laboratory, Bar Harbor, ME 04609, United States. Email: gary.churchill@jax.org.

Dietary interventions such as caloric restriction and periodic fasting improve metabolic health and extend lifespan in preclinical models, yet individuals differ widely in their physiological responses—variation that remains poorly understood but is critical for safe and effective translation to humans. We applied a 2 days per week intermittent fasting (IF) regimen to 10 inbred strains from the Collaborative Cross (CC), a genetically diverse, reproducible panel ideal for dissecting genetic effects on intervention responses. Using longitudinal phenotyping, we measured hundreds of traits, including lifespan. Our results show that sex and genetic background shape physiological responses to IF across metabolic, hematologic, and immunologic domains. Lifespan effects were also sex specific and varied among strains. These findings demonstrate that IF response is genetically determined in a mammalian model with human relevant physiology. We further compared CC results with a parallel study in Diversity Outbred mice, identifying shared predictors of health and lifespan as well as key differences between inbred and outbred populations. Overall, our work highlights the central role of genetics in shaping dietary intervention outcomes and informs efforts to translate IF benefits to human health and longevity.

Keywords: multiparental populations; gene-by-treatment interaction

## Introduction

Intermittent fasting (IF) is a dietary regimen that alternates ad libitum (AL) feeding with fasting. Common variants include time-restricted feeding, alternate-day fasting, and periodic fasting (Longo and Mattson 2014). Although IF has been less extensively studied than caloric restriction, accumulating evidence suggests that it confers geroprotective benefits (Anson et al. 2003; Carlson et al. 2007; Brandhorst et al. 2015; Xie et al. 2017; Di Francesco et al. 2018; De Cabo and Mattson 2019; Mitchell et al. 2019; Di Francesco et al. 2024).

Aging research has often implicitly generalized strain-specific findings, despite repeated evidence that genetic background modulates dietary effects on lifespan (Rikke et al. 2003; Yuan et al. 2009; Liao et al. 2010; Rikke et al. 2010; Yuan et al. 2012; Mitchell et al. 2016; Yuan et al. 2020; Mulvey et al. 2021; Roy et al. 2021; Unnikrishnan et al. 2021; Wright et al. 2022; Zhang et al. 2022). This includes our recent large longitudinal study in outbred mice, the Dietary Restriction in Diversity Outbred (DRiDO) study (Di Francesco et al. 2024). Together, these findings suggest that single-strain studies may not reliably predict dietary intervention effects in genetically heterogeneous human populations.

To investigate the potential genetic heterogeneity of IF response, we carried out the Collaborative Cross Longitudinal Study. We applied a 2-day IF intervention to 10 recombinant inbred strains from the Collaborative Cross (CC) (Churchill et al. 2004), a multiparental panel derived from eight founder strains and designed to capture reproducible genetic diversity. Longitudinal phenotyping of 800 CC mice of both sexes enabled multisystem analysis of genetic influences on dietary response across the lifespan. We conducted the CC study in parallel with the DRiDO study which included a 2-day IF intervention.

The Diversity Outbred (DO) mice are an outbred heterogeneous stock derived from the same set of founder strains as the CC. Genetic variation in dietary responses likely reflects contributions from many loci with small effects and because each DO mouse is genetically unique, the DRiDO study could not establish response heterogeneity. We compared findings across these studies to reveal common and unique features of IF response across inbred and outbred mouse populations.

## Materials and methods

### Animals

Approval for this study was obtained from The Jackson Laboratory Institutional Animal Care and Use Committee. All methods were performed in accordance with the guidelines and regulations of the Committee. The current study included 800 mice (400 females and 400 males), evenly distributed across 10 CC strains (CC003/UncJ, CC004/TauUncJ, CC005/TauUncJ, CC006/TauUncJ, CC018/UncJ, CC019/TauUncJ, CC032/GeniUncJ, CC040/TauUncJ, CC041/TauUncJ, and CC061/GeniUncJ) sourced from the Jackson Laboratory. Mice were housed in a room maintained at $70° \pm 2°F$ on a 12/12 h light/dark cycle from 6:00 AM to 6:00 PM and fed a standard chow diet (5K0G, LabDiet). Mice were observed until natural death or ethically mandated euthanasia to acquire full lifespan data. The mice lived their lifespan without handling except for cage changes every other week, scheduled feeding, and phenotyping.

### Diet

All mice were maintained on AL feeding diet until 6 mo of age. Details regarding the nutritional content of the unfasted diet are provided in the Supplementary Information. From 6 mo of age,

mice that were randomized to the AL diet had unlimited food access; fresh food was provided weekly when cages were changed. In rare instances when the AL mice consumed all food before the end of the week, the food was topped off midweek. For mice randomized to the IF diet, 48 h of fasting was imposed weekly from Wednesday noon to Friday noon. Mice on IF treatment were provided unlimited food access (similar to AL mice) on their nonfasting days. The IF regimen mimicked long periods of food deprivation that are typically experienced by animals in the wild (Anton et al. 2018). Mice were maintained on IF or AL per random assignment for the duration of their natural lifespan.

## Phenotyping

All assays were conducted at The Jackson Laboratory following standard operating procedures. Multiple assays were performed and repeated throughout the lifespan to assess physiological status. Traits were selected based on the anticipated physiological responses to IF, with lifespan as the primary outcome.

### Body weight

Body weight was assessed weekly, resulting in tens of thousands of longitudinally collected body weight values. Weekly body weights were analyzed after local polynomial regression fitting to one-third of data nearest the fitted value within mouse (ie loess smoothing).

### Metabolic phenotype

Nuclear magnetic resonance (NMR) body composition analysis was performed at approximately 12-mo intervals for each animal, providing noninvasive measurements of fat and lean mass, total body water, and free water (grams) via the Echo MRI (Houston, TX) instrument.

### Frailty and body temperature

Frailty data were collected longitudinally at 21 wk (preintervention), at 43 wk (intervention onset), and concurrent with intervention at 68, 95, 120, and 147 wks of age (ie biannually). Frailty items were scored on a two- [0,1] or three- [0,0.5,1] level ordinal scale where 0 indicated the absence of the health status deficit; 0.5 indicated mild deficit; and 1 indicated severe deficit. Temperature was concurrently collected.

### Hematologic traits

We measured hematologic traits in peripheral blood at approximately 12-mo intervals for each animal at 45, 97, and 149 wks of age. Blood samples were run on the Siemens ADVIA 2120 hematology analyzer to quantitatively measure hematologic traits.

### Immunologic traits

Peripheral blood samples were analyzed by flow cytometry using FlowJo v9.9.6 Software (BD Life Sciences) to determine the frequency of major circulating immune cell subsets. Analysis was performed before the start of dietary interventions at 5 mo, then at 16 and 24 mo of age.

### Glucose

At the flow cytometry blood collections, mice were fasted for 4 h, and glucose was measured using the OneTouch Ultra2 glucose meter from LifeScan along with OneTouch Ultra test strips.

### Lifespan

Lifespan protocols for the CC Longitudinal Study were the same as those for the DO Longitudinal Study (Di Francesco et al. 2018).

Mice were routinely evaluated for prespecified moribund criteria. If necessary, preemptive euthanasia was performed to prevent suffering; mice euthanized or found dead were classified as deaths in survival analysis.

## Statistical analysis

Initial data quality control included identifying and resolving equipment miscalibration, mislabeled animals, and technically impossible values. Quantitative assays including body weight and temperature were explored for outliers. If we could not manually correct data points using laboratory records, they were treated as missing. Mice that did not survive sample collection ($n = 2$) and/or to intervention onset at 6 mo ($n = 32$) were excluded from analyses, 21 males ($n_{AL} = 12$, $n_{IF} = 9$) and 12 females ($n_{AL} = 6$, $n_{IF} = 6$). The study was powered to detect global strain effects across the 10 inbred strains but not pairwise differences in specific strains. Results from exploratory post hoc stratified analyses within strain should be considered preliminary. Analyses were conducted using R version 4.3.3 (R Core Team 2024).

### Lifespan

We computed descriptive statistics to summarize lifespan data for each of the 10 CC inbred strains, stratified by sex and diet. Strain-specific Kaplan–Meier curves and pairwise log-rank tests estimated strain effects on lifespan. Cox proportional hazards regression models within each strain estimated the effects of treatment group, sex, and their interaction on longevity. Cox models stratified by sex and strain allowed each group its own baseline hazard. Within this framework, we compared a model with a single global diet effect across all strata to a second model estimating diet effects separately within each sex-by-strain combination. A likelihood-ratio test assess diet effect heterogeneity (GxT). Kaplan–Meier curves by treatment group and sex, along with Wald tests, were annotated with *P*-values for diet, sex, and diet-by-sex interaction effects. Nonparametric tests assessed diet effects on median lifespan within strain-sex strata. Median lifespan was estimated as the 50th percentile from survival curves using the survfit function in R's survival package. Restricted mean survival time analysis was implemented to summarize diet effects as mean lifespan difference in months via the survRM2 R package (Uno et al. 2014, 2022). To test IF's modulation of lifespan variability, we computed coefficients of variation (CV) for each strain by treatment group and conducted a Wilcox signed rank test to detect differences in CV by treatment group.

### Heritability

Genetic variation in response to dietary intervention is likely attributable to large numbers of loci with small effects. For the genetic analysis of lifespan, we assessed heritability, ie the proportion of outcome variation in a population explained by genetic relationships. Heritability can be decomposed into multiple components, such as the proportion of variation explained by all genetic effects (broad-sense heritability, or $H^2$) and the proportion of variation explained by additive genetic effects (narrow-sense heritability, or $h^2$) (Lynch and Walsh 1998). For studies of inbred strains with replicates, an intraclass correlation is an appropriate estimator of heritability, eg Yam et al. (2021). As demonstrated in Keele (2023), replicates should not be reduced to strain-level summaries for the purpose of estimating heritability due to upward bias.

Heritability estimates for lifespan in the CC mouse population were obtained using linear mixed-effects models implemented via the lme4qtl package in R, which extends lme4 to accommodate

custom covariance structures (v.0.2.2) (Ziyatdinov et al. 2018 ). Broad-sense heritability ($H^2$) was estimated using a random intercept model of the form:

$$y_{ij} = \mathbf{X}_{ij}\boldsymbol{\beta} + u_j + \varepsilon_{ij}, \tag{1}$$

where $y_{ij}$ is the survival time for replicate $i$ in strain $j$, $\mathbf{X}_{ij}$ includes fixed effects for sex, diet, and their interaction, $\mu_j \sim \mathcal{N}(0, \sigma_\mu^2)$ captures strain-specific random effects, and $\varepsilon_{ij} \sim \mathcal{N}(0, \sigma^2)$ is residual error. Narrow-sense heritability ($h^2$) was estimated by incorporating a strain-level additive genetic relationship matrix $\mathbf{K}$, derived from CC founder haplotype probabilities using the qtl2 package. A subset of the matrix (for 10 strains included in the current study) was rescaled to ensure mean diagonal equaled 1. The model was fit as:

$$y_{ij} = \mathbf{X}_{ij}\boldsymbol{\beta} + u_j, \; u \sim \mathcal{N}(0, \sigma_g^2\mathbf{K}). \tag{2}$$

Point estimate of narrow-sense heritability was extracted using the VarProp() function, and confidence intervals were computed via likelihood profiling as implemented in lme4, which is more robust than Wald-type intervals based on standard errors where estimates are bounded, as is the case for the lme4 implementation of variance components.

### Modeling strategy

We fit linear mixed models (LMMs) between outcome and study factors using the lmer function from the lme4 package in R with default control parameters. The modeling strategy can be summarized as follows:

$$\begin{aligned} \mathbf{y} = &\boldsymbol{\beta_1}(D) * \boldsymbol{\beta_2}(Se) * \boldsymbol{\beta_3}(T) + \boldsymbol{\beta_4}(Bw) \\ &+ \boldsymbol{\beta_5}(D \times Se) + \boldsymbol{\beta_6}(D \times T) + \boldsymbol{\beta_7}(Se \times T) + \boldsymbol{\beta_8}(D \times Se \times T) \\ &+ (1 + D|St) + (1|Id) + \cdots + \boldsymbol{\varepsilon}, \end{aligned} \tag{3}$$

where $\mathbf{y}$ is rank normal transformed outcome, $D$ is dietary assignment (2-day IF/ad lib), Se is sex (male/female), $T$ is scheduled time point, Bw is body weight at the date preceding and closest to 6 mo (intervention start), St is strain, and Id is a mouse-specific identifier. Batch effects not specified above (…) included additive random intercept terms for collection date (all phenotypes) and coat color and experimenter (frailty). Model complexity reflects the highly factorial nature of the study design and accounts for potential diet-by-sex-by-time point interaction, confounding by initial pretreatment body weight, gene-by-treatment interaction, clustered error variance due to replicated measurement, and batch effects. Quantification of trajectories, estimation of population-averaged intervention effect, estimation of GxT from random effects, and mouse model selection are described in detail in the Supplementary Methods.

### Comparative analysis

In a comparative analysis of IF effects across genetically diverse mouse populations (CC and DO), we conducted parallel analyses of data from the outbred DRiDO study and the inbred CC panel. We reanalyzed a subset of the DRiDO dataset, restricting to outbred mice assigned to either AL or 2-day IF protocols and applied our phenotypic response analysis pipeline to this subset to obtain comparable statistical estimates of IF effects. Thus in the current manuscript, "IF" refers to the 2-day protocol unless otherwise specified. The analysis pipeline is as described for the CC study in

Modeling strategy under Methods section, except that batch, sex, and strain were excluded (DRiDO mice were outbred females, data were preadjusted for batch). Results were compared to female-specific contrasts postestimated from LMMs in the CC dataset, as described in Supplementary Methods: Estimation of population-averaged intervention effect. The rationale for selecting traits included in the comparative analysis is also detailed in the Supplementary Methods.

### Nonlinear models of body temperature

For temperature, which was collected with greater temporal density than other phenotypes (5 time points), we reformulated time as a continuous variable by normalizing age at sampling relative to lifespan and applied generalized additive mixed model (GAMM) methods. GAMMs were fit using the mgcv R package (v.1.9-1) (Wood 2017). Random intercepts for were implemented via a smooth term with basis type "re", allowing for replicate-specific deviations. Approximate significance of smooth terms was assessed using a frequentist approximation based on F-tests. Tests are approximate in that they rely on large sample assumptions and do not fully account for the uncertainty introduced by the smoothing parameter selection process, ie the data-driven selection of effective degrees of freedom.

### Longevity biomarkers

To identify traits that are associated with lifespan, we performed regression analysis on lifespan with traits at each time point after adjusting for effects of diet, sex, strain, batch, and body weight. Weekly body weight outcome was modeled as MM and +AUC (defined above) rather than by time point. All continuous variables were rank normalized before model fitting, resulting in standardized beta coefficients. We performed likelihood ratio tests to obtain $P$-values for the adjusted associations via the anova function from the lmerTest R package (v.3.1-3) using Satterthwaite's method for computing denominator degrees of freedom and F-statistics. We applied a false discovery rate (FDR) adjustment to each test across traits and time points (one-step Benjamini–Hochberg method).

## Results

Eight hundred mice were randomized to one of two experimental arms (AL feeding [control cohort; $n = 400$] or 2-day IF [treatment cohort; $n = 400$]). Mice were weighed weekly and followed with extensive phenotyping until natural death or extreme morbidity. Mice were evenly distributed across 10 CC strains (see Materials and methods) and both sexes. At 6 mo of age, with 767 surviving mice, we initiated IF for the treatment cohort, and we maintained mice on IF or AL feeding for the duration of their natural lifespan. The experimental design leveraged sex and diverse genetic background to study variable responses to IF across hundreds of physiological outcomes collected longitudinally, including metabolic, hematologic, and immunologic health phenotypes.

### Lifespan response to IF is sexually dimorphic

We examined the effects of IF and AL on lifespan extension by sex, aggregating across the 10 inbred strains. Median lifespan was approximately 24 mo in males and 22 mo in females. Male mice tended to live longer than female mice in both treatment groups ($P = 0.038$ and $P = 1.8e{-}07$ in AL and IF, respectively). We observed a significant lifespan effect in response to 2-day IF over AL feeding among males but not females (Fig. 1a and b). Analyses pooling data from both sexes yielded comparable results among males

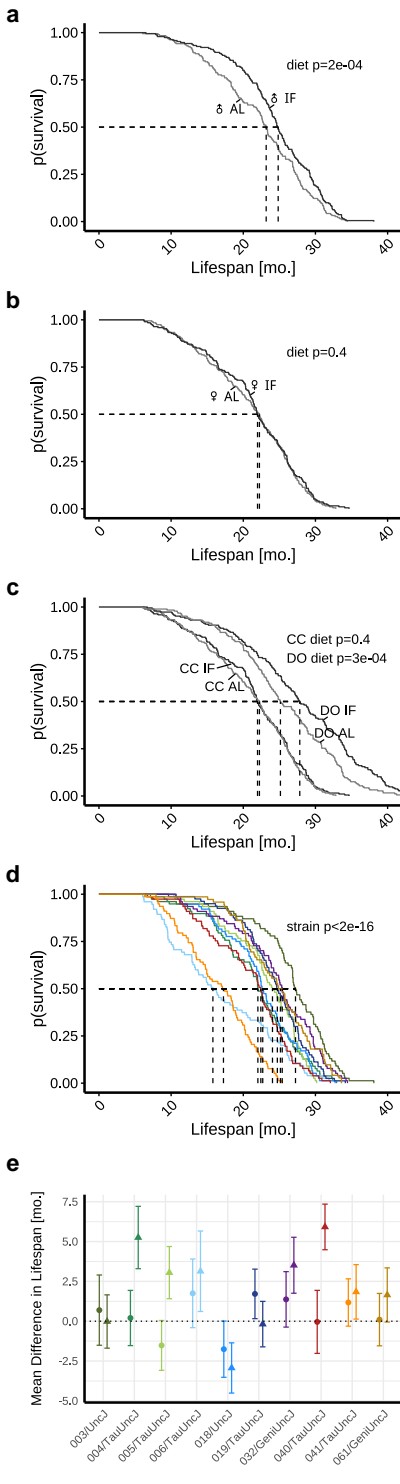

**Fig. 1.** Aggregate lifespan data demonstrates sexually dimorphic response to IF while disaggregated data by genetic background reveals interstrain variation in IF response. a, b) Study data indicate a significant lifespan extension in males subjected to 2-day IF compared to AL feeding, whereas females showed no significant response. c) Female inbred mice exhibit shorter lifespans and weaker IF responses compared to female outbred mice. Outbred data collected from the DRiDO study, which featured high genetic diversity (Churchill et al. 2004). d) Median survival varied across the 10 inbred strains under control conditions, ranging from 14.1 mo in 006/TauUncJ to 26.8 mo in 003/UncJ, underscoring the importance of genetic factors in longevity. e) Mean difference in lifespan (IF-AL) by CC strain and sex. Bars indicate ± 1 SE around the mean. CC, Collaborative Cross; DO, Diversity Outbred; mo., months.

($P = 0.00024$) and females ($P = 0.56$), as did analyses that accounted for the possibility of intracage correlations (male $P = 0.00075$, female $P = 0.78$). The lifespan extension observed in males on IF was modest (median lifespan difference = 1.7 mo) (Supplementary Table S1a). Maximum lifespan, estimated as the 90th percentile, was 30 mo in the total sample, 31.1 mo among males and 28.8 mo among females. Evidence of substantial maximum lifespan extension (>1 mo) was not observed for either sex (Supplementary Table S1b). In sensitivity analyses, we re-estimated absolute survival difference estimates of male-driven diet effects by applying restricted mean survival time (RMST) analysis (Supplementary Table S1c). The RMST difference in males was 2.02 mo (95% CI: 0.76 to 3.28, $P = 0.002$). In contrast, females showed no significant difference in survival between diets (RMST difference: 0.33 mo, 95% CI: −0.94 to 1.61, $P = 0.607$), reinforcing the conclusion that IF conferred a sex-specific survival advantage in male CC mice.

## Lifespan response to IF differs in inbred and outbred genetically diverse mice

We previously reported survival data from the DRiDO study for female DO mice including AL and 2-day IF cohorts (Di Francesco et al. 2024). The DO are derived from the same founders as the CC strains and thus share the same genetic variants. In addition, the 2 studies were carried out concurrently in the same mouse facilities. For DRiDO study methods, including information about included animals, the IF dietary intervention, and phenotyping protocols, see Di Francesco et al. (2024). We compared survival times between the 2 studies and observed that survival times of DO mice were significantly greater than the corresponding CC cohorts ($P = 1e-13$ for AL, $P < 2e-16$ for IF). This may reflect an overall fitness cost for inbreeding in the CC mice. We further noted that while IF had no effect on lifespan of female CC mice ($P = 0.39$), there was a significant effect of 2-day IF on lifespan of the female DO mice ($P = 2.6e-4$) (Fig. 1c).

## Lifespan and lifespan response to IF varies across CC strains

We observed significant strain-to-strain variation in lifespan of CC mice ($P < 2e-16$; Fig. 1d). Under AL feeding, median lifespan ranged from 14.1 mo in 006/TauUncJ to 26.8 mo in 003/UncJ, and sex-specific median lifespan ranged from 17.4 to 27.8 mo in males and 13.3 to 26.1 mo in females. Under IF feeding, median lifespan ranged from 16.9 mo in 006/TauUncJ to 28.7 mo in 003/UncJ, and sex-specific median lifespan ranged from 20 to 30.2 mo in males and 15.2 to 28.7 mo in females (Supplementary Table S2). Pairwise tests showed quantitative evidence of heterogeneity in survival by strain (Supplementary Table S3).

Significant sex-by-strain diet effects were observed as well (Fig. 1e). Using Cox models stratified by sex and strain—thereby giving each sex-by-strain group its own baseline hazard—we evaluated whether dietary effects were shared or heterogeneous across strata. Within this stratified framework, we compared a model with a single global diet coefficient to a model assigning separate diet effects to each sex-strain combination. A likelihood-ratio test supported heterogeneity of the diet effect (G×T), indicating that dietary responses differed across genetic backgrounds and sexes ($P = 0.0159$).

Within-strain, Kaplan–Meier curves and interaction contrasts for lifespan response to diet by sex further revealed interstrain variation in IF response (Supplementary Fig. S1a). Among females, five strains exhibited a reduced hazard of death on IF, while five strains showed an increased hazard, indicating both beneficial

and detrimental effects depending on genetic background. Among males, results were more consistent across strain; eight strains demonstrated reduced hazard of death on IF and only 2 strains showed increased hazards (Supplementary Table S4). Of these comparisons, statistically significant treatment effects were found only among males and all demonstrated lifespan extension on IF (diet $P = 9e-4$, 0.026, and 0.009 for males in 004/TauUncJ, 005/TauUncJ, and 040/TauUncJ, respectively). Results were similar for median lifespan extension or reduction on IF (Supplementary Fig. S1b). Strain-specific lifespan variability defined by coefficients of variation (CV; ratio of SD to the mean or SD/m) was similar between AL and IF groups, with SD around a quarter of the mean (Supplementary Table S5). Wide intrastrain variation limited power to detect strain specific IF response.

## Heritability of lifespan in CC mice

It is common in preclinical studies to test intervention effects on a single inbred mouse strain, typically C57BL/6. The current study as well as the parallel DRiDO study directly incorporate genetic diversity as feature of the study design. This approach allows for disaggregation of genotypic effects from diet effects via genetic heritability analysis. For mice that lived to at least 6 mo of age, genetic background explained 24.74% of variation in lifespan (broad-sense heritability [H2] = 0.25, 95% bootstrap CI [0.02 to 0.42]), while diet and sex explained only 3.95% of variation. We obtained genotype data and looked at the combined effects of diet, sex, and genetic relatedness as estimated by a kinship matrix. Additive genetic effects explained 26.01% of variation in lifespan (narrow-sense heritability [h2] = 0.26, 95% bootstrap CI [0.02 to 0.45]).

## Genetic background shapes physiological response to IF

Longitudinal phenotypic data collected across the lifespan were analyzed for indications that genetic background impacted IF treatment effects across multiple physiologic systems. In total, over 66,000 body weight measurements, 2,000 frailty assessments, and 1,000 each of hematology, immunology, metabolic phenotype, and glucose tolerance assays were generated by the multiyear study. We performed longitudinal analyses to identify traits significantly influenced by IF treatment as well as to determine which traits exhibited heterogeneity in response to IF by genetic background (Supplementary Fig. S2; Supplementary Tables S6–S10).

### Body weight trajectories reveal strain-specific IF responses

Weekly body weight measurements largely demonstrated canonical aging-associated trajectories, with progressive increase to peak mass at maturity followed by gradual decline postmaturity, and either rapid increase or decrease near end of life (n measurements >66 k; Fig. 2a and b). Summary metrics—mean mass (MM) and positive area under the curve (+AUC)—showed significant strain-by-diet interaction ($P = 9.96e-4$ and $2.05e-4$, respectively, Supplementary Table S9), indicating heterogeneous IF responses across genetic backgrounds (Fig. 2c). Strains with higher baseline body mass under AL feeding exhibited greater weight loss under IF (MM: $\rho = -0.78$; +AUC: $\rho = -0.58$). While summary metrics did not exhibit clear sexually dimorphic IF response at the population level (MM sex-by-diet interaction $P = 0.61$, +AUC sex-by-diet interaction $P = 0.39$), the timing of IF effects varied by sex and strain (Fig. 2a and b). For example, 004/TauUncJ male body weight responded in midlife but not late-life, and 004/TauUncJ females sustained body weight differences through late life.

### Metabolic phenotype effects are age- and strain-dependent

NMR body composition analysis provided noninvasive measurements of metabolic profiling, distinguishing lean tissue (organs, muscle) and fat mass components at 10 and 22 mo of age, roughly corresponding to middle-aged and older adult mouse life phases (n measurements >1,000). NMR-based assessments revealed that IF reduced lean mass in both sexes at 10 mo (diet $P = 8.13e-8$ in females; $P = 4.01e-8$ in males) and 22 mo (diet $P = 2.32e-6$ in females; $P = 6.01e-6$ in males) but had no significant effect on adiposity at either time point (Supplementary Table S8). Strain-specific responses were evident for lean mass (strain-by-diet variance $P = 1.21e-4$) and adiposity ($P = 2.36e-7$) (Fig. 3, Supplementary Fig. S3). Strains with greater lean mass and adiposity loss under IF tended to exhibit higher respective baseline values under AL feeding (lean mass: $\rho = -0.58$; adiposity: $\rho = -0.57$). However, 019/TauUncJ mice lost lean mass while gaining adiposity across both ages and sexes, underscoring the importance of composition-specific metrics for characterizing metabolic phenotype.

### Frailty trajectories vary by genotype but not diet

Multisystem frailty was determined by a set of 27 serially collected noninvasive biomarkers of frailty (weeks 21 [preintervention], 43 [intervention onset], 95, 121, 147; total n frailty index [FI] assessments >2,000). Per-mouse frailty trajectories were summarized as linear coefficients (slopes) on proportion of life lived (PLL [scaled age]) to describe pace of aging, and final FI score was used as a proxy for lifetime multisystem frailty accumulation. Neither the rate of frailty accumulation nor final frailty scores differed significantly between IF and AL groups (Supplementary Table S8). Further, statistical models did not support strain-specific dietary response. Simplified models without strain-specific dietary response showed strong strain effects (strain variance $P < 2e-16$ for final frailty score and $P = 9.25e-3$ for rate of frailty accumulation). Together, these results indicate genetic background modulates vulnerability to frailty, and strain-specific normative aging as measured by frailty index is unperturbed by IF in the CC panel studied.

### Lifetime incidence of select health deficits altered by IF in strain-dependent manner

Although frailty is often reported as an aggregate expression of risk resulting from accumulation of health deficits across multiple physiologic systems (Parks et al. 2012; Whitehead et al. 2014; Howlett et al. 2021), we also explored more fine-grained evidence of strain-specific aging by comparing cumulative incidence of individual health deficits (Fig. 4, Supplementary Fig. S4). Regardless of experimental condition or genetic strain, most indicators of aging were infrequently observed: breathing rate/depth, cataracts, corneal opacity, dermatitis, diarrhea, eye discharge or swelling, gait disorders, malocclusions, nasal discharge, rectal prolapse, righting reflex, tail stiffening, tremor, tumors, and genital prolapse. Alopecia and microphthalmia were also infrequently observed but there were exceptional strains. For example, CC018/UncJ was both more susceptible to alopecia in the control condition and showed stronger IF response than other strains.

Subtracting per-strain incidence among the remaining 10 indicators demonstrated varied IF response across genetic backgrounds. For instance, CC003/UncJ was particularly susceptible to loss of fur color yet showed little response to IF, and IF protocol assignment was associated with a 20% higher risk for displaying menace reflex in CC0019/TauUncJ. CC018/UncJ was vulnerable

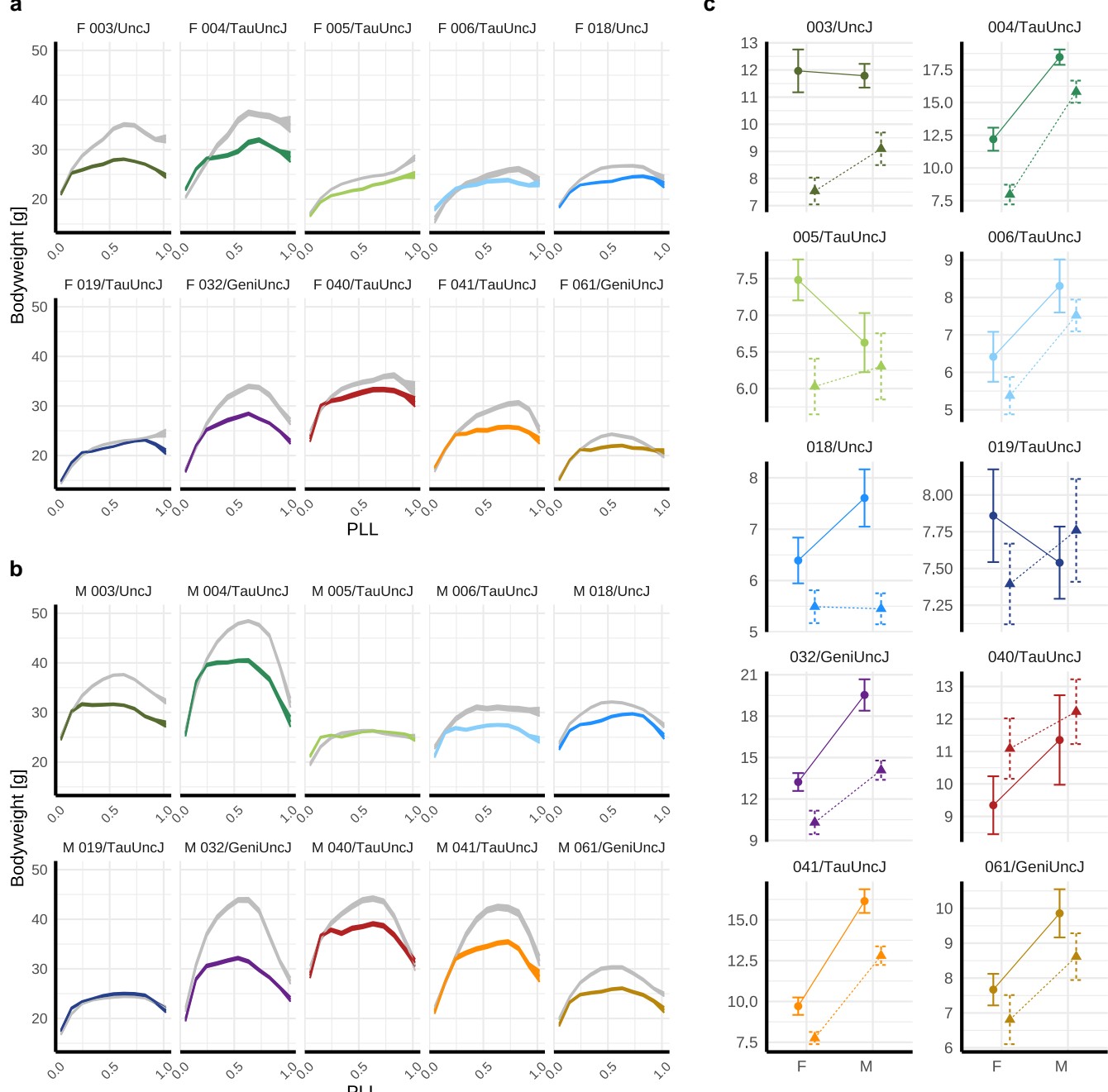

**Fig. 2.** Body weight phenotype response to IF and AL feeding across 10 inbred strains reveals sexually dimorphic genetic effects. a, b) Longitudinal body weight measurements ($n > 66$ k) were collected weekly over the lifespan of IF (color) and AL (gray) mice. Mean ± SEM body weight (g) was estimated across 20 equidistant spans of scaled age (PLL) for a) females [F] and b) males [M]. c) Longitudinal body weight measurements collected throughout lifespan were summarized per mouse as total area under the curve for body weight (+AUC[BW]; see Materials and methods) and plotted as mean ± SE +AUC[BW] for male and female mice, grouped by dietary regimen (IF [Δ] or AL [°]) and genetic strain.

to diet effects for several of the more commonly observed items. For strains with meaningful evidence of IF effect on kyphosis phenotype, kyphosis was more commonly observed in mice on IF treatment than controls, although extent of this adverse effect varied, and cumulative incidence remained below 50% for all except CC005/TauUncJ mice on treatment.

### Hematologic aging markers reveal complex effects of IF

We conducted hematophenotyping longitudinally to characterize whole blood cell type composition at 10 and 22 mo ($n$ assessments >1,000). Complete blood counts showed significantly elevated red

cell distribution width (RDW-CV) in IF-treated mice in both sexes at mid- ($P = 1.64e{-}5$ in females, $P = 8.88e{-}5$ in males) and late-life ($P = 2.1e{-}4$ in females, $P = 1.63e{-}4$ in males), consistent with erythropoietic stress. In contrast, mean corpuscular volume (MCV) did not differ significantly between IF and AL groups in either sex at either time point (Supplementary Table S8), indicating that average red blood cell size remained within normative strain- and sex-specific ranges. The combination of elevated RDW and stable MCV is consistent with early-stage or mixed anemia. This profile may reflect mild nutritional deficiencies during fasting (eg iron or B12 deficiency) or mixed anemia, in which microcytic

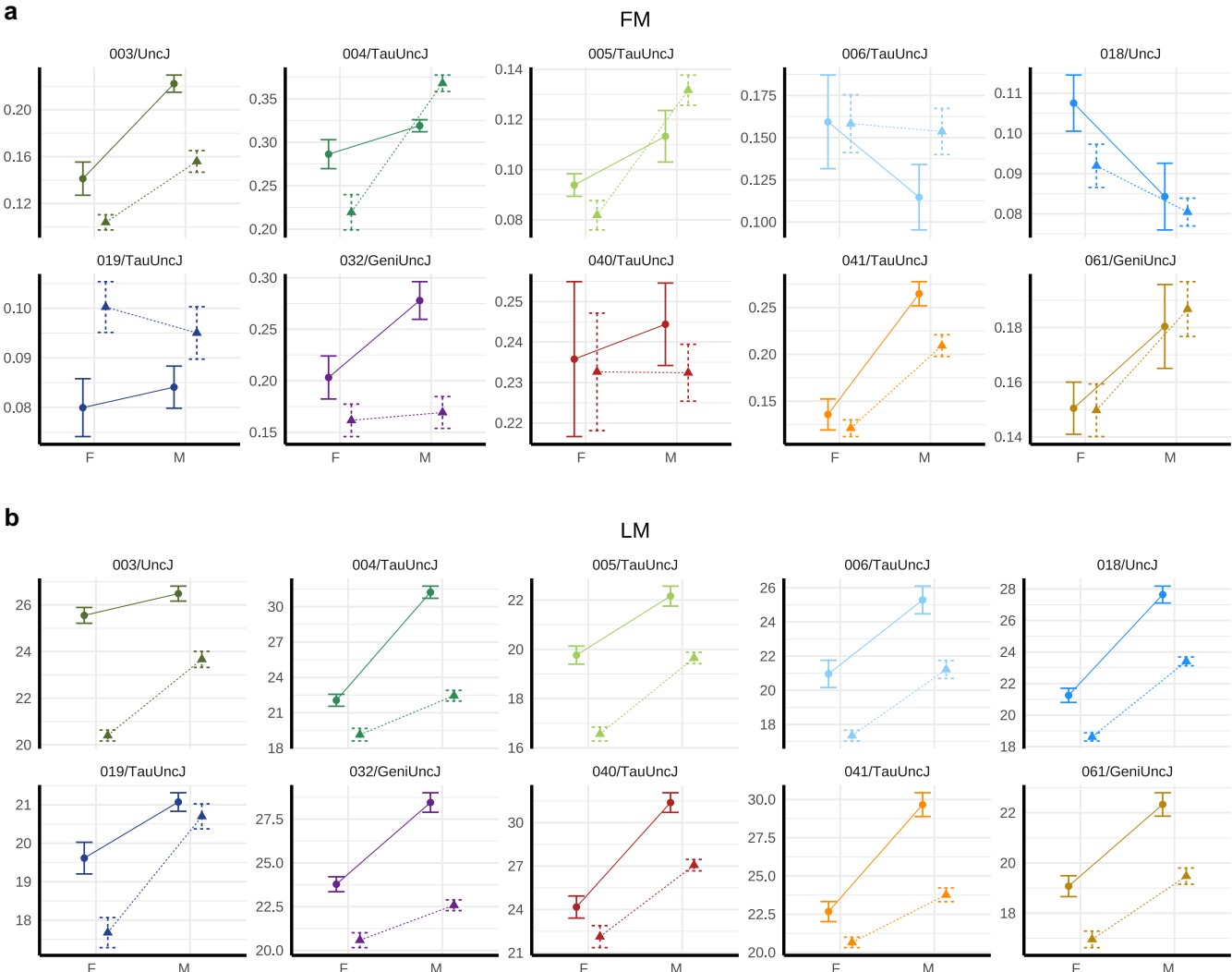

**Fig. 3.** Metabolic phenotype response to IF is influenced by genetic background. NMR body composition assays were conducted at 10 and 22 mo ($n > 1$ k). Month 10 NMR data were plotted as a) mean ± SE adiposity [%], a measure of fat mass [FM], and b) lean mass [LM, g] for male and female mice—grouped by dietary regimen (IF [Δ] or AL [°]) and genetic strain. Early lean mass loss in response to IF was observed with varied risk across strains and sexes. Some strains (eg 005/TauUncJ, 006/TauUncJ) showed minimal total mass change but significant lean mass loss, underscoring the importance of composition-specific metrics for characterizing metabolic phenotype. Plots for 22 mo shown in Supplementary Fig. S3a and b.

and macrocytic populations offset to yield a normal MCV. Strain-specific responses were evident for both RDW (strain-by-diet variance $P = 2.67e{-}18$) (Fig. 5a, Supplementary Fig. S5a) and MCV ($P = 0.0018$) (Fig. 5b, Supplementary Fig. S5b). Strains with greater RDW and MCV elevation under IF tended to exhibit lower respective baseline values under AL feeding (RDW: $\rho = -0.66$; MCV: $\rho = -0.76$).

From the broader hematologic panel, hemoglobin concentration, red blood cell (RBC) count, and reticulocyte levels are particularly informative to further characterizing the anemia-like phenotype. At 10 mo, IF was associated with sex-specific alterations: females exhibited reduced hemoglobin ($P = 0.015$) and RBC counts ($P = 0.006$); no significant differences were observed in males or older females. Reticulocyte counts were elevated in both sexes at 10 mo ($P = 4.42e{-}4$ in females, $P = 0.004$ in males) but not 22 mo (Supplementary Table S8), suggesting early compensatory erythropoietic response.

*Mouse models of hematologic response.* To further investigate the structure of hematological trait variation in response to IF

across sex and time, we visualized strain-, sex-, and time point-specific diet effect scores (see Supplementary Methods: Mouse model selection). The resulting heatmaps (Supplementary Fig. S6) revealed patterns in IF hematologic response profiles. Hematologic traits were partitioned into clusters based on IF responses in year 1 females, and row order was preserved across all panels to facilitate cross-stratum comparisons. Three major clusters emerged for hematologic trait response to IF: (i) erythroid and lymphoid morphology and abundance; (ii) granulocyte and monocyte populations alongside red cell concentration metrics; and (iii) red cell distribution and reticulocyte indices, reflecting coordinated and biologically plausible shifts in hematopoietic activity in response to IF.

Strain clusters of hematologic response profiles varied by sex and time point, with some strains showing consistent profiles across study strata and others displaying sex- or time point-specific divergence. For instance, by year 1 (midlife), four strains (003/UncJ, 004/TauUncJ, 041/TauUncJ, and 006/TauUncJ) showed similar hematologic profiles within males and within females—eg lesser vulnerability to high RDW on IF. Within males, strains

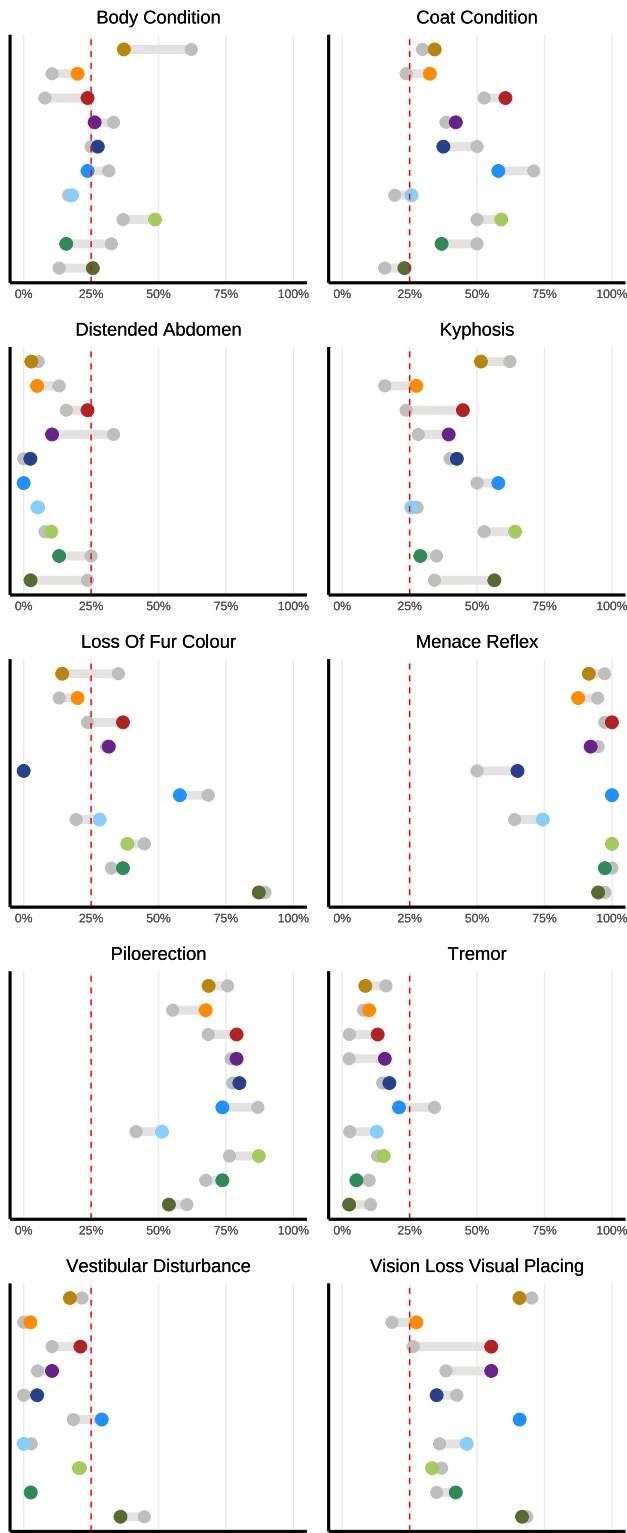

**Fig. 4.** Genetic and dietary regulation of lifetime-accumulated health deficits across multiple physiological systems. Twenty-seven serially collected noninvasive biomarkers of frailty were measured at weeks 21 (preintervention), 43 (intervention onset), 95, 121, and 147 wks ($n > 2$k). The comparison of cumulative incidence of individual health deficits across intervention groups (IF = color, AL = gray) reveals distinct aging patterns specific to each strain. Note: Remaining items shown in Supplementary Fig. S4. Frailty index items were binarized as severe/nonsevere.

032/GeniUncJ and 040/TauUncJ showed similar patterns to these four strains' midlife hematologic IF response but were distinctive from the strain/model cluster in females, suggesting mouse model development for IF response will be sex-specific. Clustering patterns were mirrored between year 1 males and year 2 females, and vice versa, further indicating complex, time-dependent strain effects on hematologic phenotypes.

## IF induces sex-specific remodeling of immune cell composition across strains

We conducted immunophenotyping longitudinally to characterize and classify various immune cell subtypes (T cells, B cells, natural killer [NK] cells, dendritic cells, and myeloid cells) and their proportions longitudinally via flow cytometry ($n$ assessments >1,000). Flow cytometry profiled major immune cell subsets preintervention at 5 (preintervention) and 16 and 28 mo (postintervention). Immune cell subsets included T cells, B cells, NK cells, dendritic cells, and myeloid cells. By 16 mo, IF elicited distinct sex-specific differences in IF response across both lymphoid and myeloid compartments, as detailed below. Low survival to the 28-mo endpoint precluded meaningful analysis.

Within the lymphoid lineage, B cell frequencies were significantly altered under IF in males ($P = 0.026$) but not in females ($P = 0.972$) at 10 mo following intervention onset. CD4$^+$ T cells also exhibited male-specific differences in IF response ($P = 0.001$), with no significant change in females (Supplementary Table S8). NK cells appeared particularly responsive to long-term dietary modulation, potentially reflecting innate immune adaptation to metabolic stress. NK cells (as a proportion of lymphocytes) were significantly altered by IF in both sexes, with a more pronounced effect in females ($P = 7.62\mathrm{e}{-6}$) than in males ($P = 1.44\mathrm{e}{-3}$). Strain-specific responses were not evident for these lymphoid lineage cells (Supplementary Table S9), suggesting low genetic modulation of IF responsiveness among lymphoid immune traits.

Within the myeloid lineage, eosinophil proportions were significantly altered in response to IF among both females ($P = 0.018$) and males ($P = 1.37\mathrm{e}{-4}$). Neutrophils, key innate responders to infection or inflammation, also showed significant response to IF in both sexes (females: $P = 1.74\mathrm{e}{-3}$; males: $P = 6.03\mathrm{e}{-5}$). In contrast, monocyte abundance (among myeloid cells) was significantly altered under IF only in males ($P = 2.27\mathrm{e}{-3}$; Supplementary Table S8). Of these myeloid subsets, monocyte abundance was the only trait to exhibit a significant strain-by-diet interaction ($P = 1.99\mathrm{e}{-3}$; Supplementary Table S9), suggesting low genetic modulation of IF responsiveness among myeloid immune traits (Fig. 5c–e).

*Mouse models of immunologic response.* To further characterize the structure of immunologic response to IF, we visualized strain-, sex-, and time point-specific diet effect scores (see Supplementary Methods: Mouse model selection). Heatmaps of IF scores (Supplementary Fig. S7) revealed patterns in strain-specific IF immune response profiles. Immunologic traits were clustered based on IF responses in year 1 females, and row order was preserved across all panels to facilitate cross-stratum comparisons. Three major clusters emerged: (i) myeloid and lymphoid cell abundance and activation (ii) a lymphocyte-dominant cluster; and (iii) a cluster dominated by inflammatory and innate immune markers, reflecting coordinated and biologically plausible shifts in immunologic activity in response to IF. Three strains (003/UncJ, 061/GeniUncJ, and 019/TauUncJ) may possess

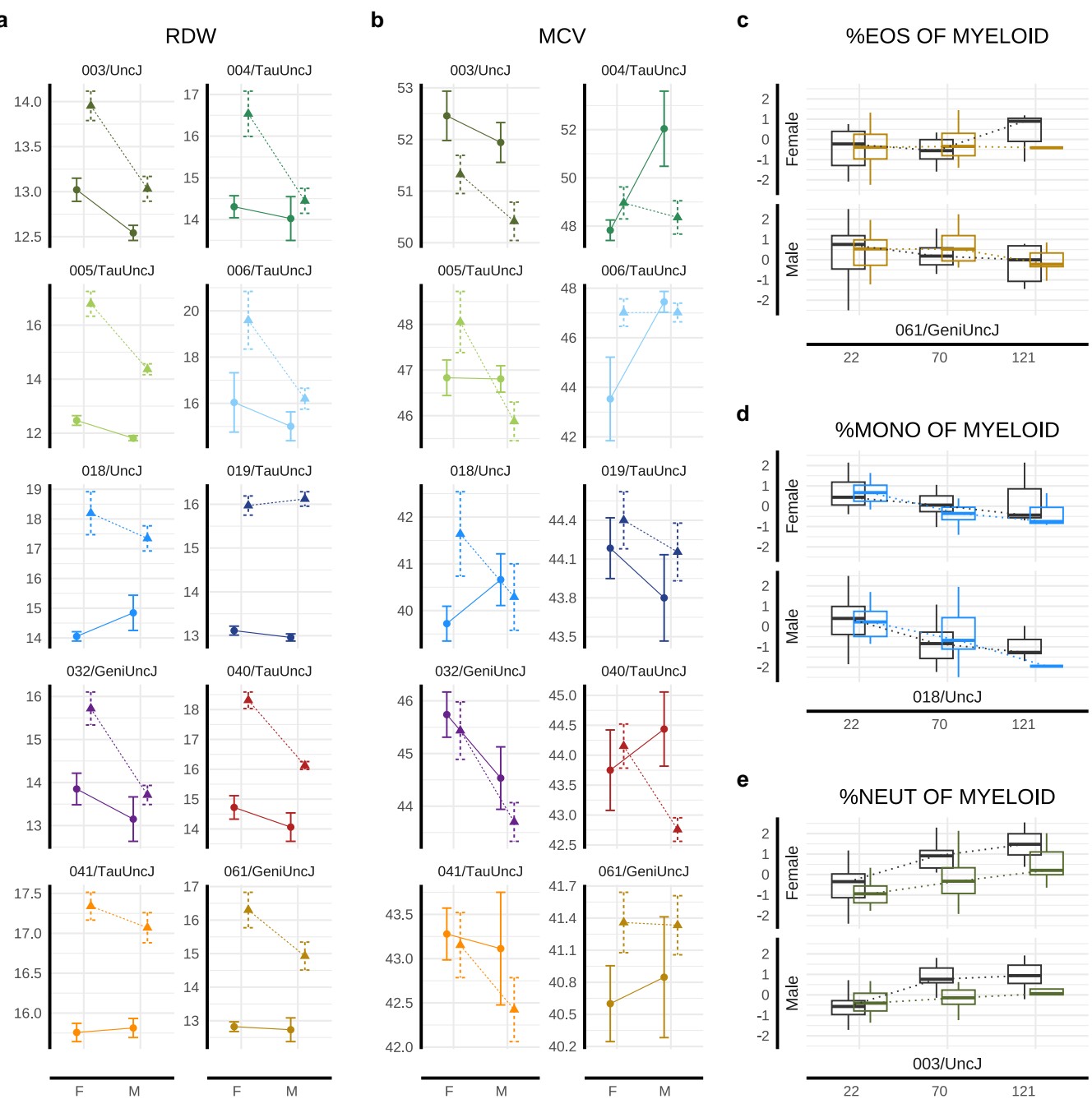

**Fig. 5.** Genetic signatures of hematologic and immunologic response to IF. Longitudinally collected data for representative hematologic and immunologic health outcomes are presented. Year 1 red blood cell distribution width (RDW CV) data (a) and mean corpuscular volume (MCV) (b) were plotted as mean ± SE for male and female mice grouped by dietary regimen (IF [Δ] or AL [°]) and genetic strain, demonstrating sexually dimorphic IF response at 10 mo. Longitudinal immunologic outcomes are presented across randomly sampled genetic backgrounds and both sexes at mid- and late-life for eosinophil (c), monocyte (d), and neutrophil/granulocyte (e) concentration among myeloid cells (IF = color, AL = gray). Strain colors are consistent across all figures and follow the key provided in (a). Of the three myeloid cell subtypes, only monocyte abundance showed statistical evidence of strain-by-diet interaction in this study. Note: 22-mo data for RDW and MCV are shown in Supplementary Fig. S5.

shared regulatory architectures governing IF-induced immune modulation, as their immune response profiles cluster together within all sex and time point strata.

## Comparative analysis finds phenotypic response to IF differs between inbred and outbred mice

We previously reported longitudinal phenotype data from the DRiDO study for female DO mice including AL and 2-day IF cohorts (Di Francesco et al. 2024). We performed longitudinal analysis in a subset of these data (mice assigned to IF or AL) to determine which traits exhibited heterogeneity in response to IF across a highly diverse outbred population (Supplementary Table S11). Despite nearly identical protocols and husbandry conditions, IF elicited both shared and divergent physiological responses in outbred (DO) and inbred (CC) mouse populations.

### Body composition trajectories diverge by genetic background

IF significantly reduced lifetime body mass in both populations (outbred +AUC P = 3.96e−14; inbred +AUC P = 3.2e−3), with comparable effect sizes. However, lean mass responses diverged:

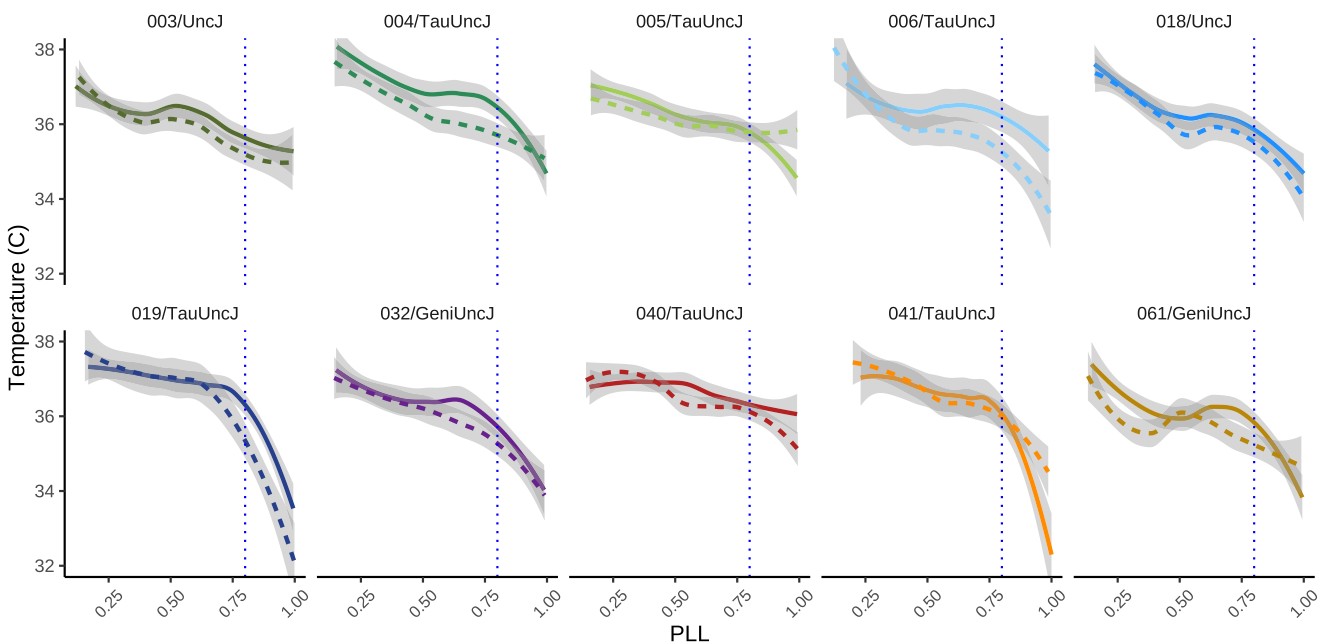

**Fig. 6.** Inflection point in body temperature at late-life transition. Body temperature plotted against PLL revealed a distinct inflection point at approximately 80% PLL for most genetic strains. This late-life shift suggests a physiological transition that may mark the onset of age-related decline or altered thermoregulatory control. Data are shown across the lifespan, with temperature values averaged within PLL bins. Note: solid line = female, dotted line = male.

outbred females maintained lean mass under IF (Supplementary Table S11), while inbred females showed significant reductions (diet $P = 8.12e–8$ at 10 mo; $P = 2.32e–6$ at 23 mo, Supplementary Table S8), suggesting strain-specific susceptibility to sarcopenia. Population-level resilience to lean mass loss among outbred females with high genetic diversity may have obscured the potential for genetic predisposition to loss of lean mass on IF observed in inbred females.

### Frailty outcomes reflect genetic context, extended lifespan in outbred mice

In outbred mice, IF reduced cumulative frailty at end of life ($P = 0.0024$) without significantly altering the rate of frailty accumulation (Supplementary Table S11), indicating improved late-life healthspan without a change in biological aging rate. In the inbred panel, frailty outcomes varied by strain but were similar across diet groups. Longer lifespan in outbred mice may have extended the window for frailty accumulation, complicating direct comparisons.

### Hematological responses to IF are highly concordant across recombinant inbred and outbred cohorts

IF increased RDW-CV, a biomarker of erythropoietic stress and aging, across both outbred and inbred populations (outbred: $P < 2e–16$ at 10 and 22 mo; inbred: $P = 1.64e–5$ at 10 mo, $P = 2.10e–4$ at 22 mo). This robust, genotype-independent effect supports RDW-CV as a candidate biomarker of adverse IF response.

### Immune cell profiles show population-specific effects

By 16 mo, IF significantly increased lymphocyte proportions (of viable cells) in outbred mice ($P = 3.15e–7$, Supplementary Table S11), whereas evidence was inconclusive in inbred females (Supplementary Table S8). IF also reduced the proportion of effector $CD4^+$ T cells (as fraction of total $CD4^+$ T cells) in outbred mice at this time point ($P = 4.89e–4$, Supplementary Table S11);

effects were not statistically significant among female inbred mice (Supplementary Table S8). The absence of significant immunologic findings in inbred female CC mice among traits selected for strong IF effects in outbred female DOs may reflect reduced genetic diversity in the recombinant panel.

### Thermoregulatory decline in late life

Generalized additive modeling was used to examine how body temperature varies with sex, strain, body weight at 6 mo, and scaled age, while accounting for repeated measures through a random effect for mouse identity. Scaled age and body weight were modeled using smooth terms to capture nonlinear effects. Temperature exhibited a well-characterized nonlinear trajectory with age, with an inflection point occurring around 80% of the lifespan, consistent with prior observations of thermoregulatory decline in late life (Fig. 6). To assess the contribution of diet, a nested model comparison was performed using an analysis of deviance. The addition of diet did not significantly improve model fit ($P = 0.386$), indicating that dietary effects are negligible in this context. Visually apparent variation in temperature patterned by sex and genetic background was not significantly associated with these factors, contrary to expectations regarding the dominant role of these intrinsic biological factors in shaping thermoregulatory dynamics.

### Longevity biomarkers

Analysis of biomarkers underlying heterogeneous longevity within intervention groups reveals several key findings (Fig. 7, Supplementary Table S12). First, greater IF-induced phenotypic changes in health and metabolic traits do not necessarily lead to lifespan extension relative to sex-, strain-, and genotype-matched peers. This observation replicates patterns from our parallel study of outbred females (Di Francesco et al. 2024) and extends these findings to inbred mice of both sexes. Second, late life adiposity was associated with higher lifespan after adjustment, indicating

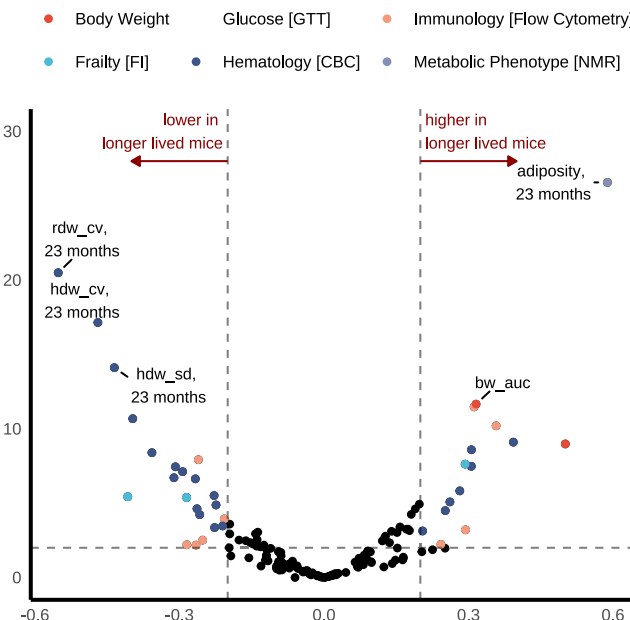

**Fig. 7.** Comprehensive longitudinal phenotyping identifies biomarkers for longevity underlying heterogeneous IF effects on lifespan within strain and sex. Regression analysis on lifespan with traits at each time point after adjusting for effects of diet, sex, strain, batch, and 6-mo body weight. All continuous variables were rank normalized before model fitting, resulting in standardized beta coefficients (x axis). We performed likelihood ratio tests to obtain P-values for the adjusted associations. FDR adjustment accounted for multiple testing. Traits are categorized and presented in volcano plots by physiologic domain. Results largely cohered with longevity biomarkers identified among female DO mice in our parallel study DRiDO. Weekly body weight outcome was modeled as MM and +AUC (defined above).

a protective longevity effect for mice that retained relatively higher proportions of adipose tissue within sex, strain, and diet group. Third, intraindividual homogeneity in red blood cells (here, hemoglobin concentration) is strongly associated with lifespan extension, more so than other hematologic traits. Fourth, both innate and adaptive immune cell types play significant roles in interindividual lifespan variation. Longer-lived mice exhibited lower levels of CD11b⁺ myeloid cells in midlife, suggesting reduced innate immune activation during this period may support longevity. In contrast, higher levels of eosinophils (as a proportion of myeloid cells) in midlife, and higher lymphocyte and B cell abundance (as proportion of viable cells and lymphocytes, respectively) in late-life, were associated with longer lifespan. These findings indicate that increased lymphocyte abundance and altered myeloid cell proportions are correlated with healthier aging at distinct life stages. Although causality remains to be determined—potentially through cell depletion studies or immune challenge models—the observed immune shifts may reflect mechanisms such as reduced inflammaging and enhanced immune memory or cancer surveillance.

## Discussion

In this study, we leveraged the genetic diversity and reproducibility of CC recombinant inbred strains to characterize heterogeneity in response to a 2-day IF intervention. We observed modest evidence for genotype-by-treatment (GxT) effects on lifespan, alongside more pronounced GxT effects on body weight, adiposity, and hematologic traits, including RDW. Together, these

findings support the hypothesis that responses to dietary interventions such as IF are genetically dependent and variable in a mammalian model with physiological relevance to humans.

Our results have several important implications (Supplementary Discussion). First, we observed sex-dependent genetic effects on multiple phenotypes, consistent with sexual dimorphism reported in large-scale mouse studies such as the International Mouse Phenotyping Consortium and the Intervention Testing Program (Karp et al. 2017; Nadon et al. 2017; Austad 2019; Sleiman et al. 2022). These patterns suggest sex-specific genetic and dietary requirements for healthy aging, effects that could not be examined in our parallel study of female outbred mice.

Second, our findings indicate broad GxT effects across metabolic, hematologic, and immunologic traits. Deviations from strain-normative phenotypes were often specific to both genetic background and sex, underscoring the importance of incorporating genetic diversity into dietary intervention studies aimed at translational relevance.

Finally, by profiling IF responses across 10 CC strains and multiple physiological systems, this study establishes a reusable resource for mechanistic exploration. These models enable replication and extension of genetically driven heterogeneity in dietary responses previously observed in outbred DO mice. Future studies can regenerate identical genotypes to confirm these findings and expand the framework to additional CC strains and their F1 hybrids, further resolving genetic contributions to variability in dietary intervention outcomes.

## Data availability

All analyses were performed using the R statistical programming language (R Core Team 2024). Data and code used to generate tables, figures, and reported results can be found on Figshare (DOI: https://doi.org/10.6084/m9.figshare.30013813). CC strain genotypes were obtained from https://www.jax.org/research-and-faculty/genetic-diversity-initiative/tools-data/diversity-outbred-reference-data. Data for DRiDO phenotypes can be found on the online data repository for the anchor manuscript (Churchill 2023).

Supplemental material available at GENETICS online.

## Acknowledgments

We acknowledge the JAX Nathan Shock Center Animal and Phenotyping Core team for their expertise in animal husbandry, data collection, and data curation, the JAX Center for Biometric Analysis for overseeing NMR data collection and processing, and the JAX Flow Cytometry Service for assistance with flow cytometry panel design, data acquisition, and gating analysis. We thank Andrew Deighan for providing assistance with data curation. DRiDO data used in this study were previously collected with support from Calico Life Sciences LLC (Dietary Intervention of Aging in Genetically Diverse Mice, sponsored research funding number CALICO-GAC-06), and are reused here with permission.

## Funding

This work was supported by the National Institute on Aging award number P30 AG038070 to Gary Churchill and Ron Korstanje (co-PI). The Jackson Laboratory Flow Cytometry Service is a Shared Resource of the Jackson Laboratory Cancer Center which is supported by the National Cancer Institute (P30 CA034196).

## Conflicts of interest

The author(s) declare no conflicts of interest.

## Author contributions

AL: methodology, formal analysis, writing. LR: project administration, data curation. WS: methodology, resources, writing. APW: formal analysis, writing. RK: funding acquisition, project administration, writing. GC: conceptualization, funding acquisition, data acquisition, project administration, methodology, formal analysis, writing.

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
