## [Peer Review File · Genetics]

Genetic regulation of fasting-induced longevity effects

Alison Luciano, Laura Robinson, William Schott, A. West, Ron Korstanje, and Gary Churchill

NOTE: The reviews and decision letters are unedited and appear as submitted by the reviewers.

In extremely rare instances and as determined by a Senior Editor or the EIC, portions of a review may be redacted. If a review is signed, the reviewer has agreed to no longer remain anonymous.

The review history appears in chronological order.

Review Timeline:

Submission Date:	2025-09-03
Editorial Decision:	2025-12-01
Revision Received:	2026-02-02
Accepted:	2026-02-04

December 1, 2025

RE: GENETICS-2025-308551

Dear Gary,

I am pleased to accept your manuscript titled "Genetic regulation of fasting-induced longevity effects" for publication in GENETICS, pending minor revision.

Please submit your revision along with a brief description of how you modified the manuscript in response to the reviewers' concerns and suggestions (which can be viewed at the bottom of this email).

There are two issues I would like you to address in your revised version. First, please improve the presentation of the experimental strategy, documenting why some phenotypes are assessed, others not, and putting the results from these phenotype assessments into the broader context of the hypothesized role of intermittent fasting in prolonging longevity (an example would be the role of RDW as a biomarker). Second, in agreement with reviewer 2, I'd like to see more attention paid to showing the size of the genotype by treatment effect, relative to other contributions to longevity. This is a key part of your paper and I think it deserves to be better presented.

I expect you should be able to submit a revised manuscript within 30 days. A suitably revised manuscript will be acceptable for publication; I don't expect to send it out for review.

When revising the ms., please make an effort to shorten it, because that almost always improves a manuscript. We urge authors to heed the advice of Strunk and White: "omit needless words"¹. Follow this link to submit the revised manuscript: Link Not Available

Thank you for submitting your work to Genetics.

Sincerely,

Jonathan Flint
Associate Editor
GENETICS

Approved by:
Anthony Long
Senior Editor
GENETICS

Reviewer comments:

Reviewer #3 :

Luciano and colleagues have examined how intermittent fasting (IF) affects lifespan and several metabolic, hematologic, and immunological traits across a set of genetically diverse inbred strains of mice. The primary question being addressed is whether genetic variation affects the response to IF. The authors performed IF for two days per week in 10 inbred strains (n = 800 mice) from the Collaborative Cross (CC). They demonstrate that sex and genetic background induce variable responses to the IF. They then compared their findings in the CC with those from a parallel study of Diversity Outbred (DO) mice. The results highlighted common predictors of health and lifespan.

The study involved a great deal of work, was performed very well, and the statistical analyses of the data were thorough. As expected, there were very significant effects of genetic background and sex on the traits studied. While there are some interesting findings (for example frailty and thermoregulation were not affected by the IF vs ad lib diets), the authors do not address the underlying mechanisms, and the fact that genetics affects the responses examined is not at all surprising. On the other hand, the results of the survey are likely to be of value for future, more detailed studies.

I confirm that the data in both the manuscript and the supplemental material support the authors' conclusions and are both available and usable.

Reviewer #4 :

This paper reports detailed analyses of 10 inbred CC strains subject to intermittent fasting or provided food ad libitum, documenting the consequences on lifespan. The authors report the consequences of fasting on numerous measures taken at multiple time points and provide estimates of heritability. They conclude that dietary treatment response is genetically dependent. Collecting these data, at this scale is a vast amount of work and deserves to be made available for investigators interested in the relationship between diet, fasting and longevity. I have the following comments.

1 The paper lacks focus. A better introduction and justification of the measures reported would make the results more accessible to readers. For example it's not explained why mouse models of hematologic response and immune cell composition would be of interest. Why are metabolic phenotype effects represented by NMR scans, rather than (or as well as) biochemical measures of metabolism? Some guidance to why certain measures are chosen, and others omitted, would be helpful.

2 The choice of strains needs to be better justified. Why these 10 strains? Can they expand the explanation, given in the limitations section, that "the 10 strains included were selected based on availability within the study's timeframe"? It is certainly a limitation but to judge how much of a limitation it would be useful to have some idea of the genetic relationships between the CC strains - how closely related are they? Are they representative of the entire cohort? Are there any genetic or other features of these ten strains that might compromise the conclusions drawn?

3 It would be helpful to compare results with those from DOI: 10.1126/science.abo3191 which reports findings from a cross derived from four classical inbred strain (UM-HET3). That paper finds that "females have a higher median life span than males (female median = 886 days, 95% confidence interval (CI) [871-897], male median = 836 days, 95% CI [816-851]), with the largest difference in the survival curves attributable to increased early mortality in males between ages of about 300 and 800 day example". By contrast, the authors here find "Median lifespan was approximately 24 months in males and 22 months in females. Male mice tended to live longer than female mice in both treatment groups ($p=0.038$ and $p=1.8e-07$ in AL and IF, respectively). We observed a significant lifespan effect in response to 2-day intermittent fasting over ad libitum feeding among males but not females (Fig. 1a, Fig. 1b)." Why should CC male mice live longer than females but UM-HET3 females live longer than males?

4 A major interest of the paper is the gene by treatment effect, yet the magnitude of this effect is rarely reported. The authors write that the "observed effect of IF on lifespan was modest relative to the within-strain variance" but this isn't well documented in the paper. The figures show strain differences, but the text lacks a summary of the overall genetic contribution to the variance.

5 One intriguing question is whether some strains show increased variance in response to the interventions: if so, for what phenotypes, and does the pattern of phenotypes tell us anything about the origin of this increased response? Are they able to say anything about the origin of non-genetic contributors to variance?

6 The section on heritability is short and only reports heritability for "mice that lived to at least 6 months of age". Did they omit estimates at different ages for a reason? How does heritability vary with treatment effect and sex?

Thank you for considering our manuscript titled "Genetic regulation of fasting-induced longevity effects" for publication in Genetics. We appreciate the positive feedback provided by the reviewers regarding the scale of data collected and the detailed nature of our analyses. All reviewer comments are carefully addressed below.

With best regards,

Alison Luciano and Gary A. Churchill

Comments highlighted by the editor

1. Explain phenotype selection.

Editor: "please improve the presentation of the experimental strategy, documenting why some phenotypes are assessed, others not, and putting the results from these phenotype assessments into the broader context of the hypothesized role of intermittent fasting in prolonging longevity (an example would be the role of RDW as a biomarker)."

Reviewer 2: "1 ...A better introduction and justification of the measures reported would make the results more accessible to readers. For example it's not explained why mouse models of hematologic response and immune cell composition would be of interest. Why are metabolic phenotype effects represented by NMR scans, rather than (or as well as) biochemical measures of metabolism? Some guidance to why certain measures are chosen, and others omitted, would be helpful."

Thank you for drawing our attention to these points. We added the following description of our trait selection process under "Phenotyping" in the Methods:

"Traits were selected based on the anticipated physiological responses to IF, with lifespan as the primary outcome."

And the following contextual background regarding the hypothesized role of intermittent fasting in prolonging longevity was added to the Supplementary Discussion. See the following in: "Concordance and divergence in the impacts of intermittent fasting on health among genetically diverse inbred and outbred mice."

"Intermittent fasting has received attention as an alternative lifespan extending intervention (Longo_2014). We elected to implement a weekly schedule with 2 consecutive days of fasting in the CC study to ensure comparability with our DRiDO study. The relative absence of lifespan response in CC mice, compared to DO mice, was a surprising outcome that limited our ability to identify responder and non-responder CC strains. In contrast, many of the metabolic and blood phenotypes displayed a robust response to IF that enabled us to demonstrate genetic heterogeneity and attribute distinct response patterns to specific CC strains."

2. Shorten the manuscript. *“When revising the ms., please make an effort to shorten it, because that almost always improves a manuscript. We urge authors to heed the advice of Strunk and White: “omit needless words.”*

Thank you for the suggestion. We moved several paragraphs from the Statistical Analysis section to a new Methods Supplement. In the Statistical Analysis section, we simplified our description of non-linear models of body temperature. We moved most of the Discussion section to a new Discussion Supplement, and removed the Conclusion paragraph, which was redundant with the Discussion section preceding it. Other sections have been lightly edited to improve clarity and reduce overall length.

3. *“...I'd like to see more attention paid to showing the size of the genotype by treatment effect, relative to other contributions to longevity.”*

Heterogeneity of IF effect on lifespan was the motivating question for the study, and we agree that these results need to be brought forward. In the original and revised manuscripts details can be found in Suppl. Fig. S1 and in Suppl Tables S2 through S4. In the revised manuscript, Methods and Results text has been updated as follows:

Methods, page 2 lines 76 – 81: “Cox models stratified by sex and strain allowed each group its own baseline hazard. Within this framework, we compared a model with a single global diet effect across all strata to a second model estimating diet effects separately within each sex-by-strain combination. A likelihood-ratio test assess diet effect heterogeneity (GxT).”

Results page 5 lines 4-12: “Using Cox models stratified by sex and strain—thereby giving each sex-by-strain group its own baseline hazard—we evaluated whether dietary effects were shared or heterogeneous across strata. Within this stratified framework, we compared a model with a single global diet coefficient to a model assigning separate diet effects to each sex-strain combination. A likelihood-ratio test supported heterogeneity of the diet effect (GxT), indicating that dietary responses differed across genetic backgrounds and sexes ($p=0.0159$).”

We have added a new panel to Figure 1 (Fig. 1e) to more clearly display the lifespan effects of IF on each sex and strain.

We note that only male mice from strains CC004, CC005, and CC040 ($p = 9.0e-4$, 0.026, and 0.009, respectively; Suppl. Table S4) reveal significant IF effects. We do not claim that IF effects are absent in other strains due to low power of the stratified analysis.

The following points from reviewer #4 were not highlighted by the editor. We briefly address them.

2 The choice of strains needs to be better justified. Why these 10 strains? Can they expand the explanation,

~~1st Accepted Version - Authors Response to Reviewers, February 2, 2026~~
given in the limitations section, that "the 10 strains included were selected based on availability within the study's timeframe"? It is certainly a limitation but to judge how much of a limitation it would be useful to have some idea of the genetic relationships between the CC strains - how closely related are they? Are they representative of the entire cohort? Are there any genetic or other features of these ten strains that might compromise the conclusions drawn?"

The reviewer is commenting on the following paragraph from the Supplementary Discussion's "Limitations" section: *"Another limitation lies in the study's sample composition. The 10 strains included were selected based on availability within the study's timeframe, constituting a convenience sample. While strain was modeled as a random effect, the representativeness of these strains relative to the broader Collaborative Cross (CC) population is uncertain. Ideally, future studies would incorporate a larger and more diverse set of strains to better capture the genetic landscape of response variability. However, scaling up the number of strains and/or the number of animals per strain poses significant logistical and resource challenges, underscoring the need for more efficient study designs in lifespan research. To fully harness the potential of multiparental recombinant inbred panels such as the CC in advancing scientific discovery, continued innovation in experimental design is needed."*

We have replaced this text with the following, which more precisely addresses the issue of strain selection: "The genetic composition of most CC strains equally represents contributions from the eight founders with no obvious allelic distortion at a per locus basis. In this sense any selection of strains would be representative of the population of (existing and potential) CC strains. However, availability of CC strains was contingent on successful breeding in the repository colony at the time of this study. Thus, we cannot exclude the possibility that these 10 strains may be atypical of the larger CC strain set in some regards."

3 It would be helpful to compare results with those from DOI: 10.1126/science.abo3191 which reports findings from a cross derived from four classical inbred strain (UM-HET3). That paper finds that "females have a higher median life span than males (female median = 886 days, 95% confidence interval (CI) [871-897], male median = 836 days, 95% CI [816-851]), with the largest difference in the survival curves attributable to increased early mortality in males between ages of about 300 and 800 day example". By contrast, the authors here find "Median lifespan was approximately 24 months in males and 22 months in females. Male mice tended to live longer than female mice in both treatment groups (p=0.038 and p=1.8e-07 in AL and IF, respectively). We observed a significant lifespan effect in response to 2-day intermittent fasting over ad libitum feeding among males but not females (Fig. 1a, Fig. 1b)." Why should CC male mice live longer than females but UM-HET3 females live longer than males?

We appreciate the relevance and importance of this publication and now cite this key reference in paragraph 2 of the discussion:

~~1st Accepted Version - Authors Response to Reviewers: February 2, 2026~~
“Our results have several important implications (Supplementary Discussion). First, they suggest sexual dimorphism of genotype effects on phenotypic traits, validating similar phenomena in large-scale mouse studies such as the International Mouse Phenotyping Consortium (IMPC) database (>14k wildtype animals⁴⁷ and >40k mutant mice) (Karp et al. 2017) and the Intervention Testing Program (Nadon et al. 2017; Austad 2019; Sleiman et al. 2022).

The reversal likely reflects genotype-by-sex interactions. CC mice are highly recombinant inbred lines with diverse founder strains, including wild-derived mice, and thus CC lines have more extreme genetic variance. Here, we hypothesize--reserving further study for future work:

- Do sex hormones interact differently with disease/aging susceptibility in the two populations?
- Are CC males less aggressive or less prone to fighting than UM-HET3 males, contributing to their lifespan advantage?
- Do CC females exhibit higher incidence of cancers or metabolic disorders due to strain-specific alleles interacting with hormone profiles?
- Or, is the sex effect reversal an experimental artifact? The ITP practices cage-wise censoring/preemptive euthanasia: “To minimize the effects of fighting on mortality, all mice in cages with overt fighting (see methods) were censored from this study and the historical studies referenced here” (Jiang 2023). Our Aging Center protocol differed such that male mice would be less likely to be censored prematurely which could increase estimated lifespan among males.

References

Jiang, N., Cheng, C. J., Gelfond, J., Strong, R., Diaz, V., & Nelson, J. F. (2023). Prepubertal castration eliminates sex differences in lifespan and growth trajectories in genetically heterogeneous mice. *Aging Cell*, 22(8), e13891.

Sleiman BM, Roy S, Gao AW, Sadler MC, Von Alvensleben GV, Li H, Sen S, Harrison DE, Nelson JF, Strong R et al. 2022. Sex- and age-dependent genetics of longevity in a heterogeneous mouse population. *Science*. 377:eabo3191.

4 A major interest of the paper is the gene by treatment effect, yet the magnitude of this effect is rarely reported. The authors write that the "observed effect of IF on lifespan was modest relative to the within-strain variance" but this isn't well documented in the paper. The figures show strain differences, but the text lacks a summary of the overall genetic contribution to the variance.

Please see Supplementary Table S5, lines 98-101 on page 2 and lines 87-92 on page 5 in the original submission where we address this points: “To test IF’s modulation of lifespan variability, we computed coefficients of variation (CV) for each strain by treatment group and conducted a Wilcox signed rank test to

~~1st Accepted Version - Authors' Response to Reviewers, February 2, 2026~~
detect differences in CV by treatment group.” “Strain-specific lifespan variability, defined by coefficients of variation (CV; ratio of standard deviation to the mean or sd/m) was similar between AL and IF groups, with standard deviation around a quarter of the mean (Supplementary Table S5). Wide intra-strain variation limited power to detect strain specific IF response.”

Also see response to Editor’s comment #3 above.

5 One intriguing question is whether some strains show increased variance in response to the interventions: if so, for what phenotypes, and does the pattern of phenotypes tell us anything about the origin of this increased response? Are they able to say anything about the origin of non-genetic contributors to variance?

Our analysis uses a different conceptual framework than the one suggested by reviewer 2’s comments. As described in lines 111-119 on page 13 and lines 1-16 on page 14 in the original submission:

“Recombinant inbred panel screens, such as the one presented here, require large sample sizes to achieve sufficient statistical power for detecting disaggregated strain-specific effects. This scale of experimentation is often impractical. Computational methods can be leveraged to maximize power to detect variability in strain-specific intervention response. In this study, we introduced a novel application of empirical Bayes estimation of strain-specific random effects to identify mouse models exhibiting heterogeneous responses to intervention. This approach conceptualizes strain in the panel screen as samples drawn from a broader population, enabling genotype-by-treatment (GxT) interactions to be modeled as a single variance component representing intervention effect variability across genetic backgrounds. The magnitude of this variance component can be compared across models of standardized phenotypes to identify traits more pronounced GxT interaction effects. Further, by leveraging empirical Bayes estimates, distinctive strains can be identified by examining the extreme values of the best linear unbiased predictors (BLUPs) for strain-specific intervention effects. E.g., selecting strains with maximum and minimum BLUPs for dietary effect enables the selection of strain pairs that exhibit divergent, and potentially directionally opposite, responses to the intervention.” [emphasis added]

Our analysis framework enabled computation of differences in mean response as a single variance component term (the extent of which is reported throughout the results for each outcome domain) and comparison of mean difference. While methods to address the challenge of estimating differentiated variance across strains, the additional analysis would be stratified and thus substantially limited by low N per strata.

This figure (left) compares the magnitude of variance difference (IF – AL) on unadjusted outcomes by strain and sex. The dotplots show the distribution of variance difference across traits within an outcome domain. Each dot indicates an outcome. A positive difference indicates an increase in variance on IF for that outcome.

Some observations: 1) body weight variance tended to decrease among females on IF and was less obviously patterned among males; 2) For several multi-outcome domains—blood, immune, and frailty traits—the diet effect on outcome variability was inconsistent for most strains. E.g., for strain 061/GeniUncJ, variance increased for some CBC outcomes and decreased for other CBC outcomes – this was true for both sexes; 3) NMR

body composition outcomes were perhaps unsurprisingly more consistent within strain in that the diet effect on variance of lean mass and adiposity was either positive or negative but rarely both. Interestingly, it appears there are strong sex effects in outcome variance response to intervention for some strains.

6 Heritability

The section on heritability is short and only reports heritability for "mice that lived to at least 6 months of age". Did they omit estimates at different ages for a reason?

We appreciate the reviewer's point and recognize that prior studies, including Jackson (2002) and Reviewer 2's paper in Science, have successfully leveraged heritability analysis across age. However, we did not include age-stratified estimates because temporal selection breaks random sampling and without fully adjusting for all known and unknown confounders—which is not feasible—the resulting estimates could inflate or distort estimates due to unmeasured confounders associated with survival. Consequently, heritability identified at older ages may reflect survival-related artifacts rather than true age-dependent genetic effects.

Reference: A. U. Jackson, A. T. Galecki, D. T. Burke, R. A. Miller, Mouse loci associated with life span exhibit sex-specific and epistatic effects. *J. Gerontol. A Biol. Sci. Med. Sci.* 57, B9–B15 (2002).

How does heritability vary with treatment effect and sex?

As suggested, we carried out heritability calculations to explore variation with treatment effect and sex. Fixed effects shift means, but they do not partition genetic variance by sex or diet. As heritability is a ratio of variance components, to study its variation, we computed heritability within each subgroup. Results (below) indicate that heritability of lifespan is broadly consistent across these subgroups.

Results: Across sex and diet subgroups, heritability estimates for survival were broadly similar, ranging from 0.23 to 0.35 with overlapping confidence intervals. This suggests that genetic contribution to variation in survival is relatively stable across these conditions. Ad libitum groups exhibited wider confidence intervals compared to intermittent fasting groups, indicating greater uncertainty or variability in genetic effects under unrestricted feeding. This pattern could reflect increase environmental heterogeneity in the ad libitum feeding condition or differential GxT interaction, warranting further investigation. This analysis is not described in the manuscript, but we provide a summary figure here.

February 3, 2026

RE: GENETICS-2025-308551R1

Dr. Gary A. Churchill
The Jackson Laboratory
N/A
600 Main St
Bar Harbor, Maine 04609

Dear Dr. Churchill:

Congratulations, your manuscript titled "Genetic regulation of fasting-induced longevity effects" is accepted for publication in GENETICS! Many thanks for submitting your research to the journal.

To Proceed to Publication:

1. Format your article according to GENETICS style: <https://academic.oup.com/genetics/pages/author-guidelines>
2. Ensure that you comply with data and community resource citation guidelines: <https://academic.oup.com/genetics/pages/author-guidelines#section-5-9-2>
3. Upload your final files at <https://genetics.msubmit.net>
4. Add oupsupport@scipris.com and genetics.oup@novatechset.com (or the domains @scipris.com and @novatechset.com) to your email program's "safe senders" list. You will be contacted by both at various points during the production process.

Notes:

- Your currently-accepted manuscript (unedited, as submitted, reviewed, and accepted) will be published at GENETICS and deposited into PubMed as an Advance Access article. Notify sourcefiles@thegsajournals.org before signing your license if you do not wish to publish your article via Advance Access.
- We invite you to submit an original color figure related to your paper for consideration as cover art. Please email your submission to the editorial office or upload it with your final files. You can submit a small-sized image for evaluation, and if selected, the final image must be a TIFF file 2513px wide by 3263px high (8.375 by 10.875 inches; resolution of 600ppi). Please avoid graphs and small type.
- After files are sent to Oxford University Press we use SciPris to manage article licensing and payment. If you do not have a SciPris account, you will receive an email from no-reply@scipris.com to sign up to use Oxford University Press' author portal. After logging in, follow the online instructions to sign your license and arrange any payment due.

If you have any questions or encounter any problems while uploading your accepted manuscript files, please email the editorial office at sourcefiles@thegsajournals.org.

Sincerely,

Jonathan Flint
Associate Editor
GENETICS

Approved by:
Anthony Long
Senior Editor
GENETICS

Review comments (if applicable):